# Large Spatial Model:
# End-to-end Unposed Images to Semantic 3D

**Zhiwen Fan**[1,2][†][*], **Jian Zhang**[3][*], **Wenyan Cong**[1], **Peihao Wang**[1], **Renjie Li**[4], **Kairun Wen**[3], **Shijie Zhou**[5],
**Achuta Kadambi**[5], **Zhangyang Wang**[1], **Danfei Xu**[2,6], **Boris Ivanovic**[2], **Marco Pavone**[2,7], **Yue Wang**[2,8]

[1]UT Austin    [2]NVIDIA Research    [3]XMU
[4]TAMU    [5]UCLA    [6]GaTech    [7]Stanford University    [8]USC

**Project Website**: `https://largespatialmodel.github.io`

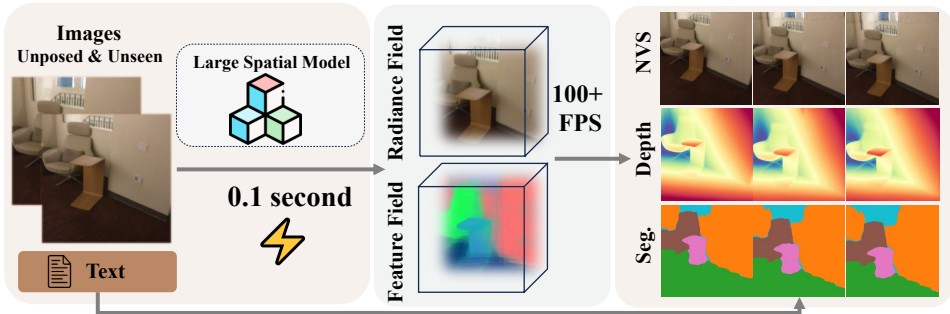

Figure 1: **Large Spatial Model** takes two unposed images as input and reconstructs an explicit radiance field, capturing geometry, appearance, and semantics in real time. This yields high performance in versatile tasks such as view synthesis, depth prediction, and open-vocabulary 3D segmentation.

## Abstract

Reconstructing and understanding 3D structures from a limited number of images is a classical problem in computer vision. Traditional approaches typically decompose this task into multiple subtasks, involving several stages of complex mappings between different data representations. For example, dense reconstruction using Structure-from-Motion (SfM) requires transforming images into key points, optimizing camera parameters, and estimating structures. Following this, accurate sparse reconstructions are necessary for further dense modeling, which is then input into task-specific neural networks. This multi-stage paradigm leads to significant processing times and engineering complexity.

In this work, we introduce the *Large Spatial Model* (**LSM**), which directly processes unposed RGB images into semantic radiance fields. LSM simultaneously estimates geometry, appearance, and semantics in a single feed-forward pass and can synthesize versatile label maps by interacting through language at novel views. Built on a general Transformer-based framework, LSM predicts global geometry via pixel-aligned point maps. To improve spatial attribute regression, we adopt local context aggregation with multi-scale fusion, enhancing the accuracy of fine local details. To address the scarcity of labeled 3D semantic data and enable natural language-driven scene manipulation, we incorporate a pre-trained 2D language-based segmentation model into a 3D-consistent semantic feature field. An efficient decoder parameterizes a set of semantic anisotropic Gaussians, allowing supervised end-to-end learning. Comprehensive experiments on various tasks demonstrate that LSM unifies multiple 3D vision tasks directly from unposed images, achieving real-time semantic 3D reconstruction for the first time.

---

[*]Z. Fan and J. Zhang contributed equally; [†] Z. Fan is the Project Lead

38th Conference on Neural Information Processing Systems (NeurIPS 2024).

# 1 Introduction

The computer vision community has devoted considerable effort to recovering and understanding 3D information (e.g., depth and semantics) from 2D sensory data (e.g., images). This process aims to derive 3D representations that encapsulate both geometric and semantic details from cheap and widely available 2D data, facilitating further interaction, reasoning, and planning within 3D physical world. Traditional approaches [1] tackle this by pipelining several distinct tasks: detecting, matching, and triangulating points for initial sparse reconstructions and the subsequent dense reconstruction, followed by the integration of specialized submodules for semantic 3D modeling.

Recent developments in this domain have markedly proceeded with a more powerful representation using both sparse reconstruction, and subsequent dense 3D modeling via Multi-View Stereo (MVS) [2, 3], Neural Radiance Field (NeRF) [4], and 3D Gaussian Splatting (3D-GS) [5], This trend influenced various industries, including autonomous driving [6], robotics [7], digital twins [8], and virtual/augmented reality (VR/AR) [9, 10]. Due to the complexity of inferring 3D information from 2D images, previous methods have broken down the holistic task into distinct, manageable subproblems. However, this strategy propagates errors from stage to stage and downgrades the performance of subsequent tasks. For instance, the critical step of precomputing camera poses -utilizing Structure from Motion (SfM) [1]— has proven to be vulnerable and often fails in scenes covered by a sparse number of views or exhibiting low-textured surfaces [11]. Such inaccuracies in camera pose estimation can ultimately lead to imprecise interpretation of the 3D scene.

Furthermore, reasoning about and interacting with the environment would benefit from a comprehensive 3D understanding. Open-vocabulary methods, which perform semantic segmentation without relying on a fixed set of labels, provide notable flexibility. However, unlike single-image understanding, the absence of large-scale and diverse 3D scene data with accurate multiview language annotations complicates the challenge. Efforts have been made to integrate 2D features into frameworks such as NeRF [12–14] and 3D-GS [15, 16]. Yet, these methods, such as Feature-3DGS [15] , typically require overfitting each 3D scene separately with extensive captured viewpoints and preprocessing camera poses using Structure-from-Motion.

To address the challenges outlined above, we propose for the first time a novel **unified framework for these key 3D vision subproblems**: *dense 3D reconstruction*, *open-vocabulary semantic segmentation*, and *novel view synthesis* from unposed and uncalibrated images. Our approach leverages a single Transformer-based model that learns the attributes of a 3D scene via *semantic anisotropic Gaussians*. Unlike previous methods that rely on epipolar Transformers with known camera parameters [17–19] or require extensive per-scene fitting [5, 15], we employ a coarse-to-fine strategy. This strategy predicts dense 3D geometry using pixel-aligned point maps, progressively refining these points into anisotropic Gaussians in a single feed-forward pass.

Our framework, dubbed ***Large Spatial Model*** (**LSM**), begins with a general Transformer architecture incorporating cross-view attention [20], which constructs pixel-aligned point maps at a normalized scale, enabling generalization across various datasets. LSM further enhances point-based representations through multi-scale fusion and local context aggregation using a ViT encoder. Additionally, LSM performs hierarchical cross-modal fusion, integrating features from a pre-trained 2D semantic model into a consistent 3D feature field. Through differentiable splatting of the regressed semantic anisotropic Gaussians, LSM enables end-to-end supervision and supports real-time scene-level 3D semantic reconstruction and rendering without needing explicit camera parameters. This allows for efficient, data-driven rendering of labels from novel viewpoints, as demonstrated in Figure 1.

Our contributions are summarized as follows:

- We introduce a unified 3D representation and an end-to-end framework that addresses dense 3D reconstruction, 3D language-based segmentation, and novel-view synthesis directly from unposed images in a single forward pass.

- Our method leverages a Transformer architecture with cross-view attention for multi-view geometry prediction, combined with hierarchical cross-modal attention to propagate geometry-rich features. We further integrate a pre-trained semantic segmentation model to enhance 3D understanding. By aggregating local context at the point level, we achieve fine-grained feature integration, enabling the prediction of anisotropic 3D Gaussians and efficient splatting for RGB, depth, and semantics.

• Our model performs multiple tasks simultaneously with real-time reconstruction and rendering on a single GPU. Experiments show that our unified approach scales effectively across different 3D vision tasks, surpassing many state-of-the-art baselines without the need for additional SfM steps.

## 2 Related Work

**SfM and Differentiable Neural Representation** Structure-from-Motion (SfM) aims to jointly estimate camera poses and reconstruct sparse 3D structures from multiple views. Traditional pipelines [1] involve multiple stages, including descriptor extraction, correspondence estimation, and incremental bundle adjustment. Recent advances in learning-based techniques [21–25] have further improved the accuracy and efficiency of SfM. These methods are widely adopted in 3D vision tasks, where differentiable neural representations typically assume accurate camera poses provided by SfM. For instance, NeRF [4] and its successors [26] rely on poses estimated offline via COLMAP [1, 27]. Similarly, 3D Gaussian Splatting [5] uses SfM-generated 3D points for initialization and has been applied to robotics [28–30], healthcare [31–33], and many other domains [34–36]. Beyond novel view synthesis, lifting 2D features to 3D has gained traction in various editing tasks [13, 15, 37, 14].

**End-to-End Image-to-3D** 3D reconstruction is a long-standing problem in computer vision, with traditional approaches like SfM [38, 39, 1], Multi-view Stereo (MVS) [3, 2, 40, 41], and Signed Distance Function (SDF) [42, 43]. More recent techniques utilize neural representations, including implicit [4] and explicit [5] formats to generate 3D models. Semantic understanding is often integrated during the reconstruction process [44], or through additional optimization steps [12, 13]. However, most methods depend on a preprocessing step like SfM [1] to estimate camera calibration, poses, and sparse point clouds before dense reconstruction, either through feed-forward prediction or test-time optimization. This reliance on calibration and pose estimation limits scalability with large-scale data, contrasting the success seen with large foundation models [45]. The latest pose-free, feedforward approaches, such as Scene Representation Transformers[46–48], have advanced the concept of representing multiple images as a "set latent scene representation," allowing for novel view generation even in the presence of inaccurate camera poses or without any pose information. However, these methods struggle to produce explicit geometry. DUSt3R[20] addresses this limitation by predicting dense point clouds directly from unposed stereo image pairs, enabling pixel-aligned geometry prediction at normalized scales. Practically, dense point prediction requires accurate multi-view RGB-D pairs, which significantly limits its scalability. InstantSplat[49] addresses this by utilizing novel-view synthesis with only posed image data and employing Gaussian Bundle Adjustment to jointly optimize camera and scene parameters for ultra-fast dense 3D reconstruction.

In contrast, our framework offers a holistic solution for **dense 3D semantic reconstruction from unposed images**. It integrates dense 3D geometry reconstruction, and language-based 3D interaction, while minimizing the need for extensive data annotation by using novel view synthesis as a core task. Since dense 3D annotations are often scarce in real-world scenarios, we propose semantic anisotropic Gaussians to lift 2D features map to 3D semantic embeddings without additional annotation. Our approach addresses higher-level 3D tasks in perception and dense 3D reconstruction compared to DUSt3R [20], by utilizing lightweight annotations and solving these tasks jointly within a unified framework.

## 3 Methods

**Overview** Figure 2 illustrates the architecture for training the Large Spatial Model (LSM). During training, the input consists of stereo image pairs along with associated camera intrinsics and poses: $\{(\mathbf{I}_i \in \mathbb{R}^{H \times W \times 3}), (\boldsymbol{T}_i \in \mathbb{R}^{3 \times 4}), (\mathbf{K}_i \in \mathbb{R}^{3 \times 3})\}_{i=1}^2$. At inference, however, unposed images can be directly fed into the framework. The pixel-aligned geometry is predicted using a standard Transformer architecture [50] with cross-attention between input views. Dense prediction heads are employed to regress normalized point maps during training: $\{\mathbf{D}_i \in \mathbb{R}^{H \times W \times 3}\}_{i=1}^2$ (see Sec. 3.1).

To support fine-grained semantic anisotropic 3D Gaussian regression, which represents the 3D scene and lifts generic feature fields from pre-trained 2D vision models, we apply point-based attention with learnable positional encoding in a local window. This propagates features from neighboring points (Sec. 3.2), effectively merging encoded features with rich semantics (Sec. 3.2) at multiple scales using 2D pre-trained models (Sec. 3.3). New views from the semantic radiance fields can be

decoded using splitting [5] on the target poses (Sec. 3.4). During inference, semantic anisotropic Gaussians are directly predicted, and the renderer takes the camera parameters derived from the point maps. An overview of the model architecture is shown in Figure 2.

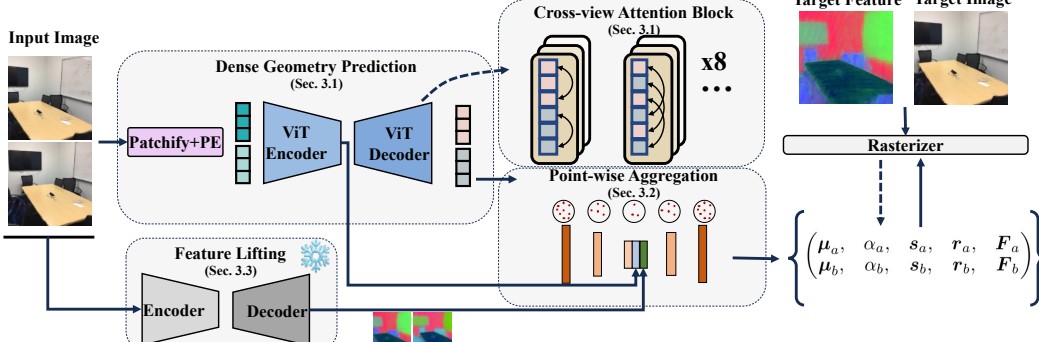

Figure 2: **Network Architecture.** Our method utilizes input images from which pixel-aligned point maps are regressed using a generic Transformer. A set of semantic anitrosopic 3D Gaussians incorporating geometry, appearance, and semantics are then predicted employing another point-based Transformer that facilitates local context aggregation and hierarchical fusion. It is supervised end-to-end, minimizing the loss function through comparisons against ground truth and rasterized label maps on new views. During the inference stage, our approach is capable of predicting the scene representation without requiring camera parameters, enabling real-time semantic 3D reconstruction.

## 3.1 Dense Geometry Prediction

Instead of adopting a conventional Transformer with Epipolar attention—which can be inefficient as pixel-wise prediction requires hundreds of queries on sampled epipolar lines [17, 18]—we implement an encoder-decoder structure for directly regressing view-specific point maps at normalized scales. Cross-view attention is utilized to aggregate multi-view information efficiently.

**Direct Regression of Normalized Depth Map**  We employ a Siamese ViT-based encoder [50] that processes stereo images using shared weights. It involves the patchification and tokenization of images, followed by the integration of sinusoidal positional embeddings. To directly regress the pixel-aligned point maps from the unposed images for view $v \in \{1, 2\}$, cross-view attention is also employed, enhancing the architecture's capacity to infer spatial relationships and propagate information between views—an approach that has proven effective in prior research [51, 20, 52]. The decoder block consists of interleaved self-attention for each view and cross-attention across views, which integrates tokens from both images. The inter-view decoder includes 12 attention blocks, akin to those utilized in previous multi-view stereo (MVS) studies [52, 20]. These blocks generate tokenized features for a subsequent Dense Prediction Transformer head (DPT) [53], which estimates a pixel-wise point map in a normalized coordinate system along with confidence value:

$$\mathcal{L}_{\text{conf}} = \sum_{v \in \{1,2\}} \sum_{i \in \mathbf{D}^v} \mathbf{M}_{v,1}^i \cdot \mathcal{L}_{\text{depth}}(v, i) - \alpha \cdot \log \mathbf{M}_{v,1}^i, \tag{1}$$

where $\mathbf{M}$ is pixel-aligned confidence map, same as DUSt3R, $\mathbf{D}$ indicates all valid points to the origin, $\mathbf{M}_{v,1}$ denotes the confidence map obtained from view $v$, expressed in the coordinate frame of view 1, $\alpha$ is a hyper-parameter that apply regularization, encouraging the network to perform robustly in challenging areas. The depth error is calculated by

$$\mathcal{L}_{\text{depth}} = \sum_{v \in \{1,2\}} \left\| \frac{1}{z} \cdot \mathbf{P}_{v,1} - \frac{1}{\hat{z}} \cdot \hat{\mathbf{P}}_{v,1} \right\|, \tag{2}$$

where the normalization factors ($z$ and $\hat{z}$) indicate that the predicted and ground-truth pointmaps are processed by normalizing. For example, $z$ is obtained by: $\text{norm}(\mathbf{P}_1, \mathbf{P}_2) = \frac{1}{|\mathbf{D}^1| + |\mathbf{D}^2|} \sum_{v \in \{1,2\}} \sum_{i \in \mathbf{D}^v} \|\mathbf{P}_v^i\|$.

## 3.2 Point-wise Feature Aggregation

Building on the foundational work in NeRF[4] and Multi-view Stereo[3], which employ a coarse-to-fine strategy for high-quality radiance field and depth estimation, we also aggregate the initial predicted geometry by applying a Transformer [54] at the point level, leveraging hierarchical representations to achieve more refined, point-based regression.

**Point-wise Attribute Prediction**  Rather than relying solely on a single network to represent the scene, we employ two Transformer-based networks optimized for distinct tasks: one for capturing "coarse" global geometry and another for "fine" local information aggregation. Initially, we integrate stereo point maps, including color information for each point primitive, formulated as $\{\mathbf{p}_i = (x_i, y_i, z_i, r_i, g_i, b_i)\}_{i=1}^N$ to serve as input. Unlike tokenized image patches, point primitives carry distinct geometric significance within Euclidean space. Inspired by recent advancements in point-cloud processing [55–57], we employ a Transformer within a localized window to perform point-wise aggregation, selectively emphasizing key features from neighboring primitives. Point-wise encoding and decoding are essential for refining scene representation, utilizing multiscale aggregation across five hierarchical levels.

After aggregating the point-wise features, we employ an additional layer of multilayer perceptron (MLP) to regress the parameters, representing the 3D scene through a set of anisotropic Gaussians [5]. The parameters include the opacity $\alpha$, scale factor $\boldsymbol{s}$, rotation $\boldsymbol{r}$, and Spherical Harmonics coefficients $\left\{\boldsymbol{c}_i \in \mathbb{R}^3 | i = 1, 2, ..., k\right\}$ where $k = (K + 1)^2$ is the number of coefficients of SH with degree $K$. The Gaussian centers $\boldsymbol{\mu}$ are regressed from geometry prediction backbone. The color $\boldsymbol{c}$ of direction $\boldsymbol{d}$ is then computed by summing up all SH basis as $\boldsymbol{c}\left(\boldsymbol{d}\right) = \sum_{i=1}^n \boldsymbol{c}_i \mathcal{B}_i\left(\boldsymbol{d}\right)$, where $\mathcal{B}_i$ is the $i^{\text{th}}$ SH basis. The final pixel intensity $\boldsymbol{c}$ is calculated by blending $n$ ordered Gaussians overlapping the pixels using the following render function:

$$\boldsymbol{c} = \sum_{i=1}^n \boldsymbol{c}_i \alpha_i \prod_{j=1}^{i-1}(1 - \alpha_j) \tag{3}$$

This equation efficiently models the contributions of each Gaussian to the pixel's final appearance, accounting for their transparency and layering order.

**Cross-model Feature Aggregation**  To effectively combine multi-view image features with point-wise geometric information, we implement cross-model attention between two sets of tokens. The attention block fuses tokens from different sources by first applying self-attention to the input $\mathbf{P}$, allowing each token to attend to other tokens within the same sequence. This process helps capture internal relationships and enrich the representation of the input token. Next, cross-attention is used, where two sets of tokens ($\mathbf{P}$ and $\mathbf{F}$) from the latent layers of two different models are fused, enabling the integration of external information into $\mathbf{P}$. Finally, a feed-forward network (MLP) further processes the updated information following cross-model fusion.

The original point features $\mathbf{P}$ contain explicit and precise spatial information, which is critical for accurate geometry reconstruction. In contrast, the image token features $\mathbf{F}$, from image encoder (Sec. 3.1) are rich in semantic content, providing important contextual information that enhances general understanding of the scene. Cross-model fusion enables the integration of detailed spatial geometry with semantic richness:

$$\mathbf{Q} = \text{Proj}(\mathbf{P}), \quad \mathbf{V}, \mathbf{K} = \text{Proj}(\mathbf{F}),$$
$$\mathbf{P} = \text{softmax}\left(\frac{\mathbf{Q}\mathbf{K}^\top}{\sqrt{d_k}}\right)\mathbf{V}$$

where $\mathbf{P}$ and $\mathbf{F}$ were normalized with a linear layer before projection.

## 3.3 Learning Hierarchical Semantics

To facilitate semantic 3D representation, we augment the anisotropic 3D Gaussians with a learnable semantic feature embedding (a.k.a. semantic anisotropic Gaussians) and rasterize into the 2D image plane by blending Gaussians that overlap with each pixel using a feature rendering function.

$$\boldsymbol{s} = \sum_{i=1}^n \boldsymbol{s}_i \alpha_i \prod_{j=1}^{i-1}(1 - \alpha_j) \tag{4}$$

$s$ indicates the final rasterized feature embedding on image plane, and $s_i$ is semantic embedding on anisotropic Gaussians.

**3D Semantic Field from 2D Images**    After obtaining $s$, we optimize $s_i$ by minimizing the difference between the rasterized feature map and the feature maps generated by a pre-trained 2D model. Unlike the previous method [15] which requires test time optimization, we transform the estimation of the feature field into a fully learnable process.

Feature maps ($\{\mathbf{S}_i \in \mathbb{R}^{H' \times W' \times N}\}_{i=1}^2$) from a pre-trained 2D multi-modal model [58] are inherently view-inconsistent due to the lack of spatial awareness during the model's training. To elevate multi-view feature embeddings into a coherent 3D feature field for holistic 3D understanding, we introduce a dynamic fusion strategy employing an attention-based correlation module. This module is specifically designed to learn blending weights for each token within Point Transformer [54] from the input pixel-wise feature embedding ($\mathbf{S}_i$). We employ attention blocks as described in Eq.3.2 to synchronize in the latent spaces through a supplementary set of cross-attention layers. The visual feature from LSeg[58], denoted as $\hat{\mathbf{S}}$, is utilized for this purpose. This loss function is minimized during training by utilizing rasterized feature maps on new views $\mathbf{S}$ and directly inferred feature maps using ground truth images on new views $\hat{\mathbf{S}}$ (LSeg [58]), thereby facilitating the learning of blending weights for consistent semantic field regression.

$$\mathcal{L}_{\text{dist}} = 1 - \text{sim}(\hat{\mathbf{S}}, \mathbf{S}) = 1 - \frac{\hat{\mathbf{S}} \cdot \mathbf{S}}{\|\hat{\mathbf{S}}\|\|\mathbf{S}\|} \tag{5}$$

**Multi-scale Feature Fusion**    To improve model efficiency, we propagate information from ViT Encoder feature $\mathbf{F}$ and the frozen semantic feature $\mathbf{S}$, to the 3D latent space (point feature $\mathbf{P}$) which has fewer tokens, thereby enabling selective attention to critical features. We further refine feature fusion across multiple stages, optimizing information flow while minimizing additional computational overhead. Novel view synthesis serves as an effective task to encode the complete geometric and appearance features into a low-dimensional 3D latent space, while recovering a set of semantic anisotropic Gaussians ($\boldsymbol{G} \in \{\boldsymbol{g}_i \in \mathbb{R}^{1 \times C}\}_{i=1}^N$) through learning from large-scale data and end-to-end training.

### 3.4  Training Objective

Putting all together, our model can be optimized end-to-end:

$$\mathcal{L} = \underbrace{\left\| \mathbf{C}(\mathbf{G}, \boldsymbol{d}) - \hat{\mathbf{C}} \right\| + \lambda_1 \cdot \text{D-SSIM}(\mathbf{C}(\mathbf{G}, \boldsymbol{d}), \hat{\mathbf{C}})}_{\text{Photometric}} \tag{6}$$

$$+ \lambda_2 \cdot \underbrace{\mathcal{L}_{\text{dist}}(\mathbf{S}(\mathbf{G}, \boldsymbol{d}), \hat{\mathbf{S}})}_{\text{Semantic}} + \sum_{v \in \{1,2\}} \lambda_3 \cdot \underbrace{\mathcal{L}_{\text{conf}}(\mathbf{D}_{v,1}, \hat{\mathbf{D}}_{v,1})}_{\text{Geometric}} \tag{7}$$

where $\mathbf{C}$ and $\hat{\mathbf{C}}$ are rasterized and GT pixel intensities, $\mathbf{G}$ denotes represented 3D scene using a set of 3D semantic anisotropic Gaussians, $\mathbf{S}$ and $\hat{\mathbf{S}}$ denotes rendered LSeg feature extractor and feature on the target image, $\boldsymbol{d}$ indicates the direction and position at new views. In our methodology, we leverage both photometric loss and semantic loss to supervise the generation of rasterized new views. In order for geometry prediction and semantic feature lifting, we employ a confidence-weighted depth loss applied to the input views. The parameters $\lambda_1, \lambda_2, \lambda_3$ are set to 0.25, 0.3, and 1.5, respectively, as determined by the grid search.

## 4  Experiments

### 4.1  Implementation Details

For our architecture, we employ ViT-Large as the encoder and ViT-Base as the decoder, complemented by a DPT head [53] for pixel-wise geometry regression. We initialize the geometry prediction layers using DUSt3R [20]. Point Transformer layers consists of 5 encoder and 4 decoder blocks with progressive downsampling and upsampling. The cross-model fusion strategy is implemented at the output of the last encoder and the output of the first decoder. The entire system is optimized

Table 1: **Quantitative Comparison in 3D Tasks.** We report novel-view synthesis, depth estimation quality, and open-vocabulary segmentation accuracy. Our method eliminates the need for any preprocessing in 3D tasks, while achieving performance comparable to other baselines that rely on SfM to obtain camera parameters and poses.

| | Reconstruction Time↓ | | Source View | | | | Target View | | | | |
|---|---|---|---|---|---|---|---|---|---|---|---|
| | SfM | Per-Scene | mIoU↑ | Acc.↑ | rel↓ | $\tau$↑ | mIoU↑ | Acc.↑ | PSNR↑ | SSIM↑ | LPIPS↓ |
| LSeg | N/A | N/A | 0.5278 | 0.7654 | - | - | 0.5281 | 0.7612 | - | - | - |
| NeRF-DFF | 20.52s | 1min2s | 0.4540 | 0.7173 | 27.68 | 9.61 | 0.4037 | 0.6755 | 19.86 | 0.6650 | 0.3629 |
| Feature-3DGS | 20.52s | 18mins36s | 0.4453 | 0.7276 | 12.95 | 21.07 | 0.4223 | 0.7174 | 24.49 | 0.8132 | 0.2293 |
| pixelSplat | 20.52s | 0.064s | - | - | - | - | - | - | 24.89 | 0.8392 | 0.1641 |
| Ours | 0.108s | | 0.5034 | 0.7740 | 3.38 | 67.77 | 0.5078 | 0.7686 | 24.39 | 0.8072 | 0.2506 |

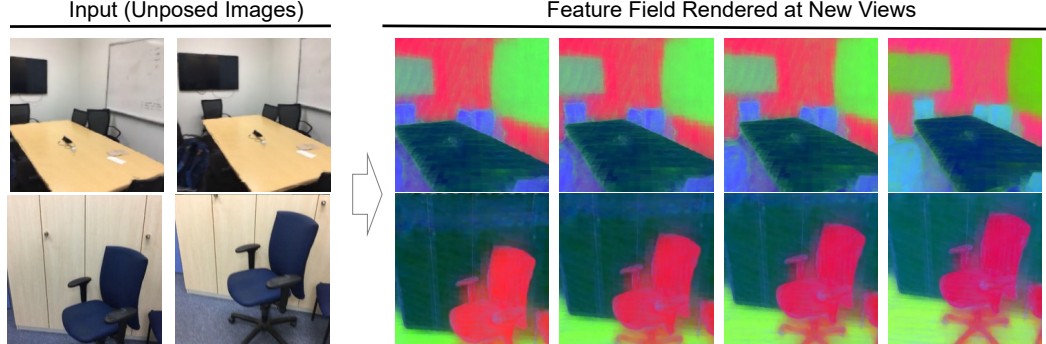

Figure 3: **Visualization of the 3D Feature Field**. We present examples of features rendered from novel viewpoints, illustrating how our method converts 2D features into a consistent 3D, facilitating versatile and efficient segmentation. Visualizations are generated using PCA [59].

end-to-end using the loss function described in Eq. 6. The training of our model contains 100 epochs, leveraging a combined dataset of ScanNet++[60] and Scannet[61], of 1565 scenes. Training is on 8 Nvidia A100 GPU lasts for 3 days. We start with a base learning rate of 1e-4 and incorporate a 10-epoch warm-up period. AdamW is employed as the optimizer for all experiments. Evaluation is conducted on 40 unseen scenes from ScanNet. Additionally, we assess on tasks: novel view synthesis, multi-view depth prediction, and 3D language-based semantic segmentation.

## 4.2 Semantic 3D Reconstruction

**Evaluation of Synthesized Images Quality**   Novel view synthesis is evaluated using NeRF-DFF [13] and Feature-3DGS [15], both of which are capable of predicting RGB values as well as features. In addition, we compared our approach with the state-of-the-art, generalizable, pose-based 3D Gaussian Splatting method, pixelSplat [18], which generates point-based representations through a feed-forward pass. Unlike our method, these existing approaches rely on known camera intrinsics and poses prior to evaluation. As indicated in Table 1, NeRF-DFF and Feature-3DGS tend to overfit on each individual scene, requiring significantly more time than our method, yet performing comparably in terms of output quality. pixelSplat utilizes an Epipolar Transformer, searching along the epipolar line using GT camera parameters to regress Gaussian attributes, resulting in longer inference times. Visualizations in Figure 4 demonstrate that our results are sharper and exhibit fewer artifacts than NeRF-DFF, and are comparable to Feature-3DGS and pixelSplat in performance.

**Evaluation of Open-vocabulary Semantic 3D Segmentation**   The semantic segmentation is evaluated by class-wise intersection over union (mIoU) and average pixel accuracy (mAcc) on novel views as metrics. Following the approach of Feature-3DGS [15], we map thousands of category labels from diverse datasets into a set of common categories, including {Wall, Floor, Ceiling, Chair, Table, Bed, Sofa, Others}. We compare our model against two state-of-the-art 3D baselines with the capacity for generating RGB, semantics and depth on any view: Feature-3DGS [15] and NeRF-DFF [13], which are based on 3D-GS [5] and NeRF [4], respectively. Additionally, the model LSeg [58], used as a 2D open-vocabulary segmenter for feature lifting, is included in our comparisons. We present statistics related to the semantic annotations on the adopted the ScanNet datasets in Table 1, where LSM demonstrates competitive performance compared to baseline 3D methods that require

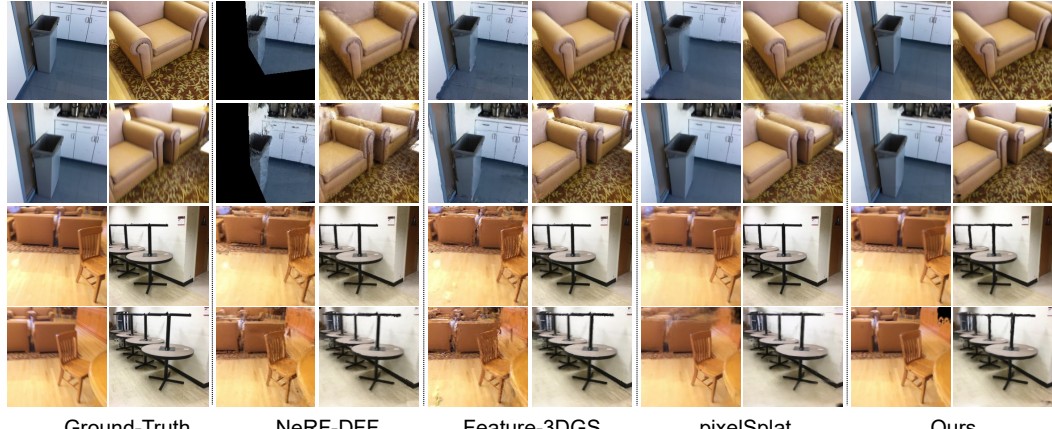

| Ground-Truth | NeRF-DFF | Feature-3DGS | pixelSplat | Ours |

Figure 4: **Novel-View Synthesis (NVS) Comparisons**. We evaluate scene-level reconstruction by comparing our method to approaches that require per-scene optimization, such as NeRF-DFF and Feature-3DGS, which predicts both RGB and segmentation, and the generalizable 3D Gaussian Splatting method (pixelSplat). Notably, these methods require a pre-processing step to obtain camera poses using off-the-shelf SfM. Through end-to-end, data-driven training, our method achieves comparable visual quality to these approaches while reconstructing the 3D radiance field in a single feed-forward pass.

Table 2: **Ablation Study on Our Design Choices.** We refer to the model that integrates cross-view attention for multi-view geometry with point-wise aggregation for future refinement as the baseline configuration (Exp #1). Implementing cross-modal attention to fuse geometry encoder features enhances both the rendering quality of new views and the segmentation accuracy (Exp #2). Additionally, incorporating features from frozen 2D semantic backbone into the fusion process (Exp #3) for consistent feature field amalgamation, and multi-scale fusion enhances hierarchical information flow (Exp #4), substantially improving language-based semantic 3D segmentation. Segmentation metrics use LSeg results as ground-truth in this table.

| Exp ID | Model | mIoU↑ | Acc.↑ | PSNR↑ | SSIM↑ |
|--------|-------|-------|-------|-------|-------|
| [1] | Baseline | 0.4562 | 0.6940 | 24.00 | 0.7981 |
| [2] | [1] + Fuse Encoder Feat. | 0.5410 | 0.8083 | 23.67 | 0.7876 |
| [3] | [1] + Fuse LSeg Feat. | 0.5586 | 0.8505 | 23.85 | 0.7902 |
| [4] | [1] + [2] + [3] + Multi-scale Fusion | **0.6042** | **0.8681** | **24.39** | **0.8072** |

ground-truth camera parameters and extensive per-scene optimization. The visualized results in Figure 5 illustrate that LSM can produce view-consistent semantic maps. In contrast, the 2D method LSeg yields detailed segmentation results but lacks cross-view consistency. To validate that LSM learns semantically meaningful features, we visualize the lifted feature field using PCA to reduce the high-dimensional features into three channels [13]. As shown in Figure 3, LSM effectively generates a faithful semantic feature field through feed-forward inference using pair images.

**Evaluation of Depth Accuracy**    We also evaluate the performance of our model on the task of multi-view stereo depth estimation. We utilize the Absolute Relative Error (rel) and Inlier Ratio ($\tau$) with a threshold of 1.03 to assess each scene, similar to DUSt3R [20]. Since our approach does not rely on any camera parameters for prediction, we align the scene scale between the predictions and the ground truth. Specifically, we normalize the predicted point maps using the median of the predicted depths and similarly normalize the ground truth depths, following procedures established in previous literature [62, 20] to align the two sets of depth maps. We observe in Table. 1 that LSM achieves state-of-the-art accuracy on ScanNet datasets than the per-scene wise methods. Our model is significantly faster than baseline methods, as it only require a forward-pass.

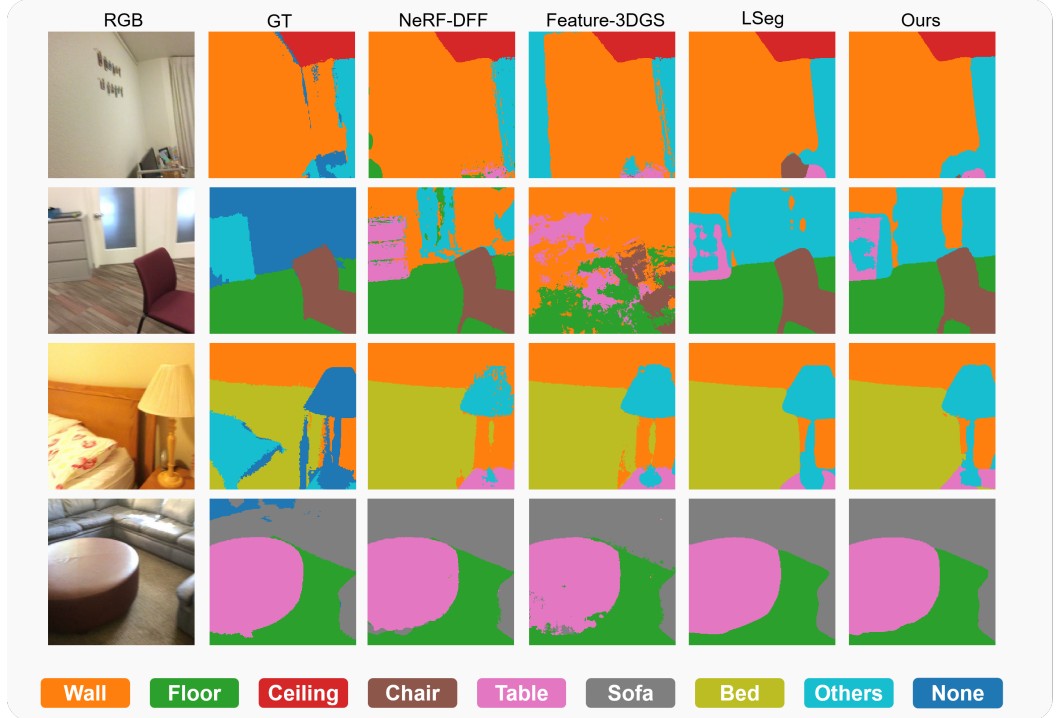

Figure 5: **Language-based 3D Segmentation Comparison**. We visualize the segmentation results across four unseen scenes and observe that our method performs comparably to NeRF-DFF and Feature-3DGS. This indicates that LSM effectively lifts 2D feature maps into high-quality 3D feature fields.

Table 3: **Inference Time per Module.** Breakdown of inference time for each module for analysis.

| Module | Inference Time (seconds) |
|---|---|
| Dense Geometry Prediction (Sec. 3.1) | 0.029 |
| Point-wise Aggregation (Sec. 3.2) | 0.046 |
| Feature Lifting (Sec. 3.3) | 0.019 |
| **Total** | **0.096** |

## 4.3 Ablation Studies

We conduct ablations to validate our desing effectiveness. Experiments are on both language-based segmentation and novel view synthesis. The quantitative results can be views at Table 2.

**Cross-Model Feature Aggregation** Incorporating the geometry encoder feature from ViT into the hidden layer of the point-aggregation layer (Sec. 3.2) demonstrates that such cross-model information flow significantly benefits the segmentation task, improving the mean Intersection over Union (mIoU) from 0.4562 to 0.5410 (Exp #1 $\rightarrow$ 2).

**Semantic Feature Fusion at Multi-Scale** Employing cross-model fusion, where latent features of the semantic model are integrated into the middle layers of point-based aggregation, also improves injection of semantically rich embeddings (0.4562 to 0.5586, Exp # 1 $\rightarrow$ 3). The decoded features confirm that the lifted feature field produces higher-quality feature maps, with the semantic mIoU improving from 0.5586 to 0.6042 (Exp #3 $\rightarrow$ 4) through multi-scale fusion.

**Module Timing.** We analyze the computational cost of each module by running inference 1,000 times on the ScanNet test dataset with the model, as shown in Table 3, and calculating the average inference time for each module of the Large Spatial Model.

Table 4: **Performance Comparison on Replica Dataset.** LSM operates without ground-truth camera parameters, achieving decent PSNR and low relative depth error, while also enabling semantic understanding within a unified framework.

|  | Model | Offline SfM | mIoU ↑ | rel ↓ | PSNR ↑ |
|---|---|---|---|---|---|
|  | pixelSplat | Required | None | 20.14 | 26.28 |
| Replica | Splatter Image | Required | None | None | 12.37 |
|  | Ours | Not Required | 0.51 | 4.91 | 23.10 |

**Evaluation of Generalizable Methods on New Datasets.** To avoid potential overfitting, we adopt the Replica dataset[63], a photorealistic simulated 3D dataset with accurate RGB, dense depth maps, and semantic annotations for comprehensive evaluation. We use the same data preparation with Feature-3DGS. LSM generalizes well to the simulated Replica test set, achieving the best depth estimation metrics and enabling 3D semantic segmentation, which is unique among generalizable methods. Splatter Image[64], an ultra-fast monocular 3D object reconstruction method using 3D-GS, performs well for object reconstruction with masked backgrounds but struggles with scene-wise reconstruction in complex backgrounds.

## 5    Conclusion, Limitation, and Broader Impact

We have introduced the Large Spatial Model (LSM), a unified framework for holistic 3D semantic reconstruction from uncalibrated and unposed images, with the added capability of interaction through language. LSM leverages cross-view attention to aggregate multi-view cues and utilizes multi-scale cross-modal attention to integrate semantically rich features into a point-based representation. Hierarchical point-wise aggregation layers further refine these representations and enhance the integration of cross-modal attention. By splatting regressed anisotropic 3D Gaussians, LSM enables the generation of novel views with versatile label maps. LSM is highly efficient, capable of real-time end-to-end 3D modeling, and supports various downstream applications.

While our method significantly accelerates semantic 3D scene reconstruction, it relies on a pre-trained model for feature lifting, which can increase GPU memory requirements during training, especially when the integrated 2D model has a large number of parameters. Additionally, the need for ground-truth depth maps, although there are millions of multi-view datasets annotated with them, could limit its scalability for internet-scale video applications.

Our research enables efficient, real-time 3D scene-level reconstruction and understanding, which is advantageous for applications such as end-to-end robotic learning, AR/VR, and digital twins. However, there is potential for misuse, such as the arbitrary distribution of digital assets or privacy leakage related to building structures. These risks can be mitigated by embedding watermarks into the 3D assets [65].

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

## Technical Appendices

We have included visualizations of rendered new views, visualized features, and the final language-based segmentation videos can be seen from our webpage.

**Training/Testing Split.** Similar to NeRF literatures, we select one image out of four as test images, and the rest ones used as training for Feauture-3DGS and NeRF-DFF. For pixelSplat and ours, we directly use the rest ones as source-view images to reconstruct the 3D representation. We use the last checkpoint for evaluation.

**How to Derive Camera Parameters from Normalized Point Maps.** We obtain pixel-aligned point map at where we can build the mapping from 2D to the camera coordinate system. We can first solve the simple optimization problem based on the Weiszfeld algorithm [66] to calculate per-camera focal, the same as DUSt3R [20]:

$$f^* = \arg\min_f \sum_{i=0}^{W} \sum_{j=0}^{H} \boldsymbol{O}^{i,j} \left\| (i', j') - f \frac{(\boldsymbol{P}^{i,j,0}, \boldsymbol{P}^{i,j,1})}{\boldsymbol{P}^{i,j,2}} \right\| \tag{8}$$

where $i' = i - \frac{W}{2}$ and $j' = j - \frac{H}{2}$ denote centered pixel indices. Assuming a single-camera setup similar to that used in COLMAP for a single scene capture, we propose stabilizing the estimated focal length by averaging across all training views: $\bar{f} = \frac{1}{N} \sum_{i=1}^{N} f_i^*$ The resulting $\bar{f}$ represents the computed focal length that is utilized in subsequent processes. Relative transformation $\{\boldsymbol{T} = [\boldsymbol{R}|\boldsymbol{t}]\}$ can be computed by RANSAC [67] with PnP [68, 69] for each image pair.

**Additional Model Details.** We utilize the initial geometry prediction from DUSt3R, which provides pixel-aligned geometry as the starting point. The subsequent point-wise aggregation is implemented using Point Transformer V3 [54]. The 2D-trained model, LSeg, is employed to provide multi-modal feature embeddings through its tokenization module, using the feature from the second-to-last layer of the DPT head. Additionally, the last layer of the ViT encoder is integrated into the feature space of Point Transformer. The fusion is carried out by a single standard attention block, facilitating cross-model information flow. We will release the code. The middle two layers of the Point Aggregation Module are utilized for this fusion. Both Feature-3DGS and NeRF-DFF models are trained with 5,000 iterations to prevent overfitting on real-world outward-facing scenes, and they also lift features from LSeg for the creation of a 3D feature field. The point-wise aggregation module consists of four encoder blocks with progressive downsampling, and four decoder blocks with upsampling operators. The depth of each block is configured as $\{1, 1, 1, 1\}$ for the encoders and $\{1, 1, 1, 1\}$ for the decoders, respectively.

