# OpenReview forum: "Large Spatial Model: End-to-end Unposed Images to Semantic 3D"
_NeurIPS.cc/2024/Conference — NeurIPS 2024 poster_

### Official Review · Reviewer_8Lhg · 2024-07-10

**Soundness:** 3
**Presentation:** 3
**Contribution:** 3
**Rating:** 7
**Confidence:** 4

**Summary:**

The authors proposed the Large Scene Model (LSM), a novel 3D scene understanding framework that unifies multiple vision tasks within a single model. LSM represents a scene using pixel-aligned point maps, integrating geometric, appearance, and semantic information into a unified representation. By leveraging a Transformer architecture with cross-view and cross-modal attention, the model effectively incorporates multi-view cues and semantic knowledge from a 2D vision model.

LSM's design enables efficient scene-level 3D semantic reconstruction and rendering in real time on a single GPU. The model's integration of a 2D semantic model allows for open-vocabulary understanding, extending its applicability to diverse real-world scenarios. Furthermore, by consolidating multiple tasks within a single model, LSM minimizes error propagation, leading to more robust and accurate results compared to state-of-the-art baselines.

**Strengths:**

* The described technical approach in this work is sound and clearly presented. The contributions from the various proposed modules are well ablated and investigated in the experiments (Table 4).
* The model demonstrates high inference efficiency compared to other approaches. with reconstruction time of 0.1s and rendering at 270 due to the underlying 3DGS representation that is being generated.
* I like that the model reconstructs the underlying 3D representation in a single feedforward pass, as compared to multiview + test time optimization for fusion approaches. This improves the speed and efficiency for inference. It is good to see compelling quality based on the novel view synthesis.

**Weaknesses:**

* I think the main contribution of this paper is the unification of the various scene modeling tasks into the same model, including geometry, color and semantics. The authors further claimed in the abstract and introduction that multitask training end-to-end allows LSM to outperform state-of-the-art baselines. However the paper did not ablate the multi-task learning design choice. For instance, what if some of the tasks are removed (e.g., semantic feature prediction). How does that affect the performance of the other tasks?
* A suggestion is that for Figure 5, it is unclear how much pose divergence there is between the input source view and the synthesized novel view. It would be helpful to also show the source view supplied as input to the model.
* The paper is named Large Scene Model, which seems to suggest something to do with model parameter scaling, hence large. However the paper does seem to do much scaling on model size. So perhaps a more accurate terminology would be Multitask or Unified Scene Model?

Nits.
* Line 153: Typo: to serve as input?
* In Tables 1-4, I suggest highlighting the best (and possibly second-best result) for easier comparison of the various experiments.
* In Table 4, why is + Multi-scale Fusion indented?

**Questions:**

- For the quantitative comparisons given in Figure 3, are they predicted from the input view, or are they for a novel view?

**Limitations:**

- My understanding is that in this method, all the Gaussians being generated are pixel aligned with the original input images. Is that a limitation of the method, since that would make the model unable to model large pose divergences that require rendering regions not originally visible in the input view, for instance, the back side of a sofa etc.

---

> ### Author Rebuttal · Authors · 2024-08-07
>
> We thank reviewer 4 (8Lhg) for recognizing the contribution of our paper and offering insightful comments. Please find our response to the feedback below.
>
> **[W1]: Ablate the multi-task design choice?**
> We ablate the “novel view feature synthesis (Eq.4)” and “geometry prediction (Eq.1)” tasks in the following table. The row labeled “no Feat.” indicates the removal of the multi-scale fusion of LSeg features (green part in Point-wise Aggregation of Fig. 2). We observe that incorporating semantic task learning into point-wise aggregation enables the model to lift view-inconsistent feature maps to a view-consistent feature field, while it does not impact view synthesis and depth tasks a lot. We believe the input to “Point-wise Aggregation” contains pixel-wise point maps (x, y, z) and RGB (r, g, b) from the support images, which already provide the complete observation within the input views for interpolating novel views and depth prediction. Thus, additional incorporation of feature maps does not provide visual cues to enrich the radiance field.
> Removing geometric supervision leads to the model diverging quickly, which means pixel-aligned depth regression helps the model recognize correspondences across input views, guiding the model's learning when it is unaware of the camera parameters. The experiment setting follows Table 4 of the main draft.
>
>
> | Methods/Metrics           | mIoU ↑ | Acc. ↑ | rel ↓  | &tau; ↑ | PSNR ↑ | SSIM ↑ |
> |---------------------------|--------|--------|--------|---------|--------|--------|
> | Full Model                | 0.599  | 0.8125 | 4.09 | 61.39 | 21.10  | 0.7752 |
> | Full Model (no Feat.)     | None     | None  | 4.18 | 60.84 | 20.97  | 0.7745 |
>
>
> **[W2]: Update Figure 5, and the pose divergence for the novel views?**
> We will include the source (support) views in the revision; thank you for the suggestion. We specify the rendered novel camera positions and orientations by interpolating between the first and second reference (or support) images for free-view exploration (see the supplementary webpage). Specifically, the orientation interpolation is conducted using Spherical Linear Interpolation [1] between the two sets of quaternions, while the position interpolation is conducted using linear interpolation.
>
> **[W3]: More accurate terminology?**
> We will take your suggestion into account and revise it accordingly.
>
>
>
> **[Q1]: Typos and highlighting the table?**
> We will fix it.
>
>
> **[Q2]: Why is “+ Multi-scale Fusion” indented in Table 4?**
> Indented row (Multi-scale Fusion) means it is based on the rows above (Fuse LSeg Feat. and Fuse Encoder Feat.), indicating that we apply multi-scale fusion to both. We have clarified this part in the revision.
>
>
> **[Q3]:  Input view or novel view in Figure 3?**
> They are novel views.
>
> **[Limitation]**
> Thanks for pointing this out. Our method, along with all the adopted methods (pixelSplat, Feature-3DGS, NeRF-DFF), belongs to the category of reconstruction models, which focus on building the 3D based on input observations. A discussion about extending to generation for synthesizing new viewpoints with large pose divergences will be included in our revision.
>
>
> **Reference:**
> [1]. Slerp from wikipedia (we cannot provide the direct link in author response)

---

> > ### Comment · Reviewer_8Lhg · 2024-08-12
> >
> > Thank you for your clarifications and further ablations on LSeg features and thanks for considering my suggestions for adding the source view and revise the "Large Scene Model" terminology. Overall I will maintain my rating after checking notes with the other reviewers.

---

> > > ### Author Response · Authors · 2024-08-13
> > > **Responses to Reviewer's Comment**
> > >
> > > We are grateful for your positive recommendation and the supportive feedback on our paper.

---

### Official Review · Reviewer_MCB6 · 2024-07-12

**Soundness:** 3
**Presentation:** 2
**Contribution:** 3
**Rating:** 5
**Confidence:** 4

**Summary:**

This paper presents the Large Scene Model (LSM), which generates semantic radiance fields from uncalibrated RGB images using a unified Transformer-based framework. LSM can infer geometry, appearance, and semantics simultaneously and synthesize label maps in real-time. The model integrates multi-scale fusion and features from 2D models to enhance accuracy and efficiency.

**Strengths:**

1. Unified Framework: LSM combines multiple 3D vision tasks into a single framework, streamlining the process and reducing complexity.

2. Real-time Performance: The model achieves real-time 3D reconstruction and rendering, suitable for applications needing fast processing.

3. Enhanced Feature Fusion: By incorporating 2D model features, LSM improves the quality of feature lifting and semantic understanding, enhancing overall performance.

**Weaknesses:**

1. Dataset: I recommend the authors organize the training and testing phases in alignment with previous methods (NeRF-DFF and Feature-3DGS) and provide results on the Replica Dataset. The authors have not sufficiently justified deviating from the baseline evaluation split. Furthermore, an explanation is needed for the significant performance discrepancy of the baselines between the Replica Dataset and the authors' setup. Additional training details may also be necessary.

2. Writing: The paper's abstract, introduction, and methods sections require improvement.  Specifically, the methods section should introduce each module and their interconnections from a high-level perspective rather than presenting them as isolated components.

3. Method Details: Do the authors use camera parameters? If so, why are camera parameters mentioned in line 117? If camera parameters are used, the model cannot be described as "unposed."

4. Visualization: In Figure 4, there are category colors that are not listed in the legend. Additionally, a more diverse set of results should be displayed, as the current experimental set predominantly features sofas.

**Questions:**

1. Module Timing: I am curious about how the authors manage to use eight cross-attention modules and still achieve reconstruction in 0.1 seconds. Please provide the time consumption for each step.

2. Image Resolution: What is the resolution of the images? More details regarding the inference process should be provided, especially concerning the time comparison.

**Limitations:**

The authors have discussed limitations.

---

> ### Author Rebuttal · Authors · 2024-08-07
>
> We thank reviewer 3 (MCB6) for recognizing the contribution of our paper and offering insightful comments. Please find our response to the feedback below.
> **[W1.1]: Reasons for deviation from Replica, and performance discrepancy.**
> The reasons for the deviation are threefold:
>
> 1). The processed Replica datasets (by Feature-3DGS) lack ground-truth depth maps for geometric accuracy. Feature-3DGS uses COLMAP for SfM results, leading to misaligned coordinate systems.
> 2). Feature-3DGS uses LSeg as GT for evaluating semantic quality, while we use annotated GT semantic labels.
> 3). Replica datasets are simulated under ideal environments, whereas our draft focuses on real-world textures, lighting changes, and sensor noise.
>
> The performance discrepancy of Feature-3DGS stems from resolution and semantic ground-truth differences. Our experiments follow the generalizable NVS method, pixelSplat, using 256x256 resolution and cropped boundaries, reducing the valid field-of-view. We also use annotated ScanNet for semantic segmentation, rather than LSeg as ground-truth.
>
> **[W1.2]: Results on Replica Dataset with explanation.**
> Comparing our methods with NeRF-DFF and Feature-3DGS is challenging due to their assumptions of dense views, long optimization steps, and precomputed camera poses, while ours uses sparse and pose-free views. pixelSplat also assumes known camera parameters. This places our method at a disadvantage, but Tab. 1 in our submission demonstrates its strength.
>
> To further validate, we compare with Feature-3DGS on the Replica dataset, using Semantic-NeRF [1] preprocessed data with annotated RGB-D-Semantics (256x256 resolution). We randomly sample 2,000 points from GT depth maps for initializing Feature-3DGS and use 70 images for training and 10 for testing, following their setup.
>
> LSM generalizes well from the real-world ScanNet test set (25.44dB) to the simulated Replica test set (23.10dB) in zero-shot, even without GT camera parameters. Our model builds the feature field in a feed-forward pass, achieving high quality. Notably, our method achieves the best geometric accuracy without using epipolar attention (pixelSplat, need pose) or requiring test-time optimization with dense views and GT camera parameters (Feature-3DGS, need pose).
>
>
>
> | Methods/Metrics  | Support View | GT Camera Parameters | Training Time | mIoU (Seg.) ↑ | rel (Depth) ↓ | PSNR (RGB) ↑ |
> |------------------|---------------|----------------------|---------------|---------------|---------------|--------------|
> | Feature-3DGS     | 70            | Required             | 18.5 min      | 0.62          | 8.25          | 31.89        |
> | Nerf-DFF         | 70            | Required             | 1.1 min       | 0.49          | 14.33         | 24.67        |
> | pixelSplat       | 2             | Required             | 0.06 sec      | None          | 20.14         | 26.28        |
> | Ours             | 2             | Not Required         | 0.09 sec      | 0.51          | 4.91          | 23.10        |
>
>
>
> To be more aligned with our assumed setting, which are more practical in real applications, we proposed to benchmark the performance of investigated baselines under the sparse view setting. We reduce the support view number for all methods from 70 to 20, while preserving the same test sets. In the table below, Feature-3DGS drastically decrease the PSNR with degraded geometry and semantic accuracy.
>
> | Methods/Metrics  | Support View | GT Camera Parameters | Training Time | mIoU (Seg.) ↑ | rel (Depth) ↓ | PSNR (RGB) ↑ |
> |------------------|---------------|----------------------|---------------|---------------|---------------|--------------|
> | Feature-3DGS     | 20            | Required             | 18.5 min      | 0.46          | 32.44         | 19.61        |
> | Nerf-DFF         | 20            | Required             | 1.1 min       | 0.39          | 45.09         | 16.27        |
> | pixelSplat       | 2             | Required             | 0.06 sec      | None          | 27.13         | 20.62        |
> | Ours             | 2             | Not Required         | 0.09 sec      | 0.45          | 5.614         | 18.80        |
>
> **[W1.3]. Training details.**
> We included training details in Sec. 4.1. We use the default configuration to train Feature-3DGS and NeRF-DFF. For pixelSplat, we use its pretrained checkpoint and code without any modifications to perform the evaluation.
>
>
>
>
> **[W2]:Detailed module design.**
> We will add a diagram to illustrate module interconnections in the revision. Additionally, we plan to release the source code for reproduction.
>
>
> **[W3]:Why use camera parameters in L117?**
> They are needed in training and evaluation to obtain the ground-truth point maps, and specify the target view. They are not required for inference.
> **[Q1]: Additionally, a more diverse set of results should be displayed.**
> Please refer to supplementary webpage for ten free-view videos. We also incorporate additional comparisons in the Fig.3 of the attached PDF.
>
>
> **[Q2]:Module Timing? Image resolution?**
> We study the computational cost (time) of each module by inferring 1,000 times with the model and obtaining the average inference cost for each model.
> | Module | Inference Time (seconds) |
> |--------|--------------------------|
> | Dense Geometry Prediction (Sec.3.1) | 0.0297               |
> | Point-wise Aggregation (Sec.3.2)   | 0.0464               |
> | Feature Lifting (Sec.3.3)   | 0.0199               |
> | **Total** | **0.096**        |
>
> The tested resolution (256x256) is aligned with the generalizable 3D-GS method pixelSplat.  pixelSplat is slightly faster than our framework 0.06s while it requires additional steps to run Structure-from-Motion (e.g., 20.52s for each scene on ScanNet on average) to obtain the camera parameters.

---

> > ### Comment · Reviewer_MCB6 · 2024-08-13
> >
> > Thanks for your great efforts! After reading the response, some major issues have been addressed well, so I still lean towards positive for the submission. I encourage the author to add these clarifications to the main paper.

---

> > > ### Author Response · Authors · 2024-08-13
> > > **Responses to Reviewer's Comment**
> > >
> > > Thank you very much for your positive feedback and for recognizing our efforts in addressing the major concerns. We are glad that our clarifications have been helpful, and we will certainly incorporate them into the main paper as you suggested.
> > >
> > > We are always **open to further suggestions or feedback** that could help us improve the paper even more. If there are no additional concerns, we would greatly appreciate it if you could kindly consider **raising the rating**.
> > >
> > > Thank you again for your valuable input.

---

### Official Review · Reviewer_fir7 · 2024-07-13

**Soundness:** 3
**Presentation:** 3
**Contribution:** 3
**Rating:** 6
**Confidence:** 4

**Summary:**

The paper aims to train a network that takes in a set of unposed images and directly produces a semantic radiance field.

The method utilizes a single Transformer-based model that learns the attributes
of a 3D scene represented by a point-based radiance field.  A decoder produced 3D Gausians that can be splatted to make novel images, depth estimates, and semantic segmentations.

**Strengths:**

The paper provides a transformer architecture for producing 3D gaussians with rich features from unposed images, which seem very valuable.   The design choices in the proposed system are well-chosen from methods available at this time, leading to a system that has a good combination of little-compute and competitive-accuracy on three different tasks (nvs, depth, semantics).

**Weaknesses:**

The paper shares goals and ideas with "Scene Representation Transformers" (Sajjadi et al., CVPR 2022) and its follow up work Object Scene Representation Transformer (NeurIPS 2022) and RUST: Really Unposed SRT (CVPR 2023).   This paper is different, because it ultimate produces a set of gaussians rather than a LLFF or NeRF volume, and it distills features from 2D foundation models.   However, it is similar in that a transform encoder and decoder produces a scene representation directly from a set of images, which is then used for novel view synthesis, depth estimation, and semantic segmentation.   In any case, those paper seem fairly similar in concept and so I think they should be discussed in the related work, and possibly approach sections.

The ablation study in table 4 suggests that the key methods in the paper have little impact on the results of NVS.

**Questions:**

It is interesting that the 3D methods, which have access to multiple views of the same scene do not perform as well as LSeg in Table 1.   This is counter-intuitive.  Can you please explain why?

The results on multiview depth accuracy are kinda amazing.   Why is the proposed method better than ones that take the camera parameters?   Is it due aligning the scene scale (do all the methods get the same method for scale alignment?

The novel view synthesis images look very good.   Can you please provide some info about how close the novel cameras are to the reference ones provided at inference time?   Is there a way for you to quantify and compare to PixelSplat the NVS results as the novel cameras deviate further and further from the reference ones?

**Limitations:**

Limitations are discussed briefly in the second paragraph of the conclusion.

---

> ### Author Rebuttal · Authors · 2024-08-07
>
> We thank reviewer 2 (fir7) for recognizing the contribution of our paper and offering insightful comments. Please find our response to the feedback below.
>
> **[W1] Discussion w/ SRT and RUST.**
> Scene Representation Transformer (SRT)[1] and RUST[3] have pioneered the exploration of representing multiple images as a "set latent scene representation" (OSRT[2] utilized Slot Scene Representation) and generating novel views even in the presence of flawed camera poses or without any pose information. In contrast, our method targets a holistic model for semantic 3D reconstruction from unposed images, using point-based representation for efficient rendering speed. We will include this discussion in the revision.
>
> **[W2] Table 4 suggests little impact on the results of NVS.**
> To formulate the well-represented radiance field, the Point-wise Aggregation module takes the pixel-aligned (x, y, z, r, g, b) as input (line 153) for regressing point-wise attributes, which already contain the complete appearance observation within the input views for interpolation. Thus, additional incorporation of feature maps does not provide visual cues to enrich the radiance field. However, the Point-wise Aggregation module design is important as it enables the lifting of features from inconsistent stereo image features to consistent feature fields, and the Multi-scale Fusion further improves the lifting accuracy.
>
> **[Q1] Novel view semantic segmentation with LSeg?**
> We notice that LSeg fails to produce view-consistent segmentation results but generates reasonably good per-view segmentation results. Our hypothesis is that LSeg is a well-trained 2D model for language-driven semantic segmentation, serving as a foundation for our method to understand 3D. Our method lifts the view-inconsistent multi-view feature maps from LSeg into a consistent 3D feature field in a zero-shot manner, utilizing the Point-wise Aggregation module. See the figures in the attached PDF (Fig.2) to see the consistency difference.
>
> **[Q2] High multiview depth accuracy induced by aligning scene scale?**
>
> Yes, the improved geometry accuracy stems from the alignment and combination of scenes, and our training data consists of dense depth annotations of various scenes for supervision.
>
> **[Q3] How to define novel cameras? Can they deviate from the reference one?**
> Thank you for acknowledging the good rendering quality. The rendered novel camera positions and orientations are determined by interpolation between the first and second reference (or support) images. Specifically, orientation interpolation is conducted using Spherical Linear Interpolation [4] between the two sets of quaternions, while position interpolation is conducted using linear interpolation.
> We acknowledge that both our method and pixelSplat are reconstruction methods that build the radiance field using pixel-aligned Gaussians for interpolation, and thus cannot perform generation tasks that significantly deviate from the reference images.
>
>
>
> **Reference:**
> [1]. Scene Representation Transformer: Geometry-Free Novel View Synthesis Through Set-Latent Scene Representations, CVPR 2022
> [2]. Object Scene Representation Transformer, NeurIPS 2022
> [3]. Rust: Latent neural scene representations from unposed imagery, CVPR 2023
> [4]. Slerp from wikipedia page  (we cannot provide the direct link in author response)

---

> > ### Comment · Reviewer_fir7 · 2024-08-13
> > **Response to rebuttal**
> >
> > Thanks for responding to my questions in the rebuttal.   The answers are pretty-much as expected.
> > I will leave my review score unchanged.

---

> > > ### Author Response · Authors · 2024-08-13
> > > **Responses to Reviewer's Comment**
> > >
> > > We are grateful for your positive recommendation and the supportive feedback on our paper.

---

### Official Review · Reviewer_K6pm · 2024-07-14

**Soundness:** 1
**Presentation:** 2
**Contribution:** 3
**Rating:** 5
**Confidence:** 3

**Summary:**

This paper solves the sparse-view scene reconstruction problem by Large Scene Model, a unified scene reconstruction model via unposed RGB images. The model utilizes a ViT backbone for extracting the feature and uses cross-view attention to align the multi-pose feature for consistent features. The 3D scene is further rendered from the 3D semantic field derived by the multi-view features. The unified model is capable of multiple 3D-based tasks including novel view synthesis and 3D language-based segmentation. Experiments showed that the work achieves better results with limited performance sacrifices in the NVS task and higher performance in the multi-view language-based segmentation task.

**Strengths:**

1. The model is general and multi-purpose in sparse-view scene reconstruction.
2. The model can achieve better results while still obtaining lighting-fast rendering speed and can be applied to real-time reconstruction.

**Weaknesses:**

1. The technical contribution is limited. The model is generally designed via multi-purpose modules glued attention and Transformers, which is a straightforward and widely applied idea. There is no significant new problem has arisen and novel solutions proposed.
2. The performance comparison with NVS-related works is limited. Firstly, the authors train and run comparison experiments on the same dataset, which can be biased. Secondly, several popular scene datasets incorporated in similar works (such as RealEstate10k) are not utilized in this work. Thirdly, methods similar to pixelSplat such as Splatter Image[1] are not included in comparison.
3. The presentation can still be improved. Firstly, the authors titled their work “Large Scene Model”, while the design is more similar to the idea of pixel-based Gaussian splatting (such as pixelSplat and GaussianImage). Secondly, each module's input and output data type cannot be directly recognized from the pipeline graph.
4. The bibliography of this paper lacks some related works, such as Splatter Image[1], which is also an image-space Gaussian splatting method.

Reference:
[1] Szymanowicz, Stanislaw, Chrisitian Rupprecht, and Andrea Vedaldi. "Splatter image: Ultra-fast single-view 3d reconstruction." In Proceedings of the IEEE/CVF Conference on Computer Vision and Pattern Recognition, pp. 10208-10217. 2024.

**Questions:**

1. Did the authors try replacing the designed module with large-scale pretrained models, such as using pretrained monocular depth estimation model?
2. The design philosophy is similar to multi-view image generation works. Can this model output high-quality and consistent multi-view images, as the fashion of Free3D?
3. The term of “language-driven segmentation” is not quite clear to me. Does it mean semantic segmentation?

**Limitations:**

The authors claimed that the major drawback of the model is the VRAM consuming. The social impact of this work mainly related to potential misuse of 3D assets and can be solved via integrating watermarks into the generated result.

---

> ### Author Rebuttal · Authors · 2024-08-07
>
> We thank reviewer 1 (K6pm) for recognizing the contribution of our paper and offering insightful comments. Please find our response to the feedback below.
>
> **[W1]: No significant new problem has arisen and novel solutions proposed?**
> We acknowledge that our work builds upon the contributions of many giants, such as DUSt3R [1] (end-to-end point map regression model), pixelNeRF [2] (one of the first generalizable NeRFs), pixelSplat [3] (one of the first generalizable 3D-GSs), and the concept of large-scale training. However, we respectfully disagree with the assertion that "no significant new problem has arisen and novel solutions proposed."
> We target a very practical yet under-explored scenario: end-to-end semantic 3D reconstruction directly from images. While previous literature reduces this problem to “Computing Camera Parameters” followed by “Training 3D Representation”. We, for the first time, unify these problems into one differentiable vision system.
> Technically, while there are many alternative design choices, our method successfully integrates these components into an end-to-end trainable system using a standard Transformer architecture, demonstrating the feasibility of future scalable 3D training.
> We kindly suggest that the methods and ideas presented in this paper carry significant research value that can impact future 3D deep learning studies.
>
>
>
>
>
>
> **[W2]: Evaluation of generalizable methods on new datasets.**
> To avoid any possibility of overfitting with existing generalizable methods and considering the suggestion by Reviewer #MCB6, we adopt the Replica dataset [3], which is a photorealistic simulated 3D dataset containing accurate annotations of RGB, dense depth maps, and semantic maps for thorough evaluation. Specifically, the LSM generalizes well from the real-world ScanNet test set (25.44dB) to the simulated Replica test set (23.10dB). LSM also produces the best depth estimation metrics and is the only generalizable method that can enable 3D semantic segmentation. Splatter Image is an ultra-fast monocular 3D object reconstruction method using 3DGS, and we utilize the provided checkpoint for evaluation. However, while Splatter Image addresses object 3D reconstruction well when the background is masked out, it cannot handle scene-wise reconstruction well with complex backgrounds (please see the attached Fig.1).
>
> | Methods/Metrics    | GT Camera Parameters | mIoU ↑ | rel ↓ | PSNR ↑ |
> |--------------------|----------------------|--------|-------|--------|
> | pixelSplat         | Required             | None     | 20.14 | 26.28  |
> | Splatter Image     | Required             | None      | None      | 12.37  |
> | Ours               | Not Required         | 0.51 | 4.91 | 23.10  |
>
>
>
>
> **[W3]: Improve the presentation.**
> We will highlight the input and output formats in the methodology section in a future version.
> Although our method architecture exhibits some similarity to generalizable Gaussian models, we emphasize that the investigated problem is very different. pixelSplat focuses on the standard NVS setting, where **calibrated images** are assumed and the emphasis is solely on image synthesis. In contrast, our overarching goal is to develop a versatile 3D foundation model that unifies scene reconstruction and understanding with **uncalibrated images**, which is more aligned with real-world applications to deploy a single model. We will adjust the title to emphasize our unification of 3D tasks according to Reviewer 8Lhg's suggestions. Thank you for the feedback.
>
> **[W4] Missing citation.**
> Splatter Image [4] presents one of the first solutions for ultra-fast monocular 3D object reconstruction using 3D Gaussian Splatting as a representation. It takes a single or two object-wise images as input for reconstructing the 3D representation without test-time optimization, while demonstrating promising reconstruction accuracy. We will incorporate a detailed discussion in our revision.
>
>
> **[Q1] Replace encoder using monocular depth?**
> The “Dense Geometry Prediction” cannot be replaced with a monocular depth estimator. Our method takes unposed images as input to build global 3D point maps with versatile attributes for many 3D problems. It requires a stereo depth estimator to unify the point maps from two viewpoints into the same coordinate system.
>
>
> **[Q2] Can this model perform generation, like Free3D?**
> Our work reconstructs 3D scenes based on unposed images, similar to 3D-GS, but without the need for COLMAP pre-computation. The supervision comes from the new view interpolated between the training views, meaning it is not capable of extrapolating viewpoints (generation). We will clarify this point and discuss the potential to extend the proposed framework into a generative approach in the future work section.
>
> **[Q3] What’s “language-driven segmentation”?**
> "language-driven segmentation" means segmenting areas based on a set of language descriptions  (e.g., “wall,” “floor”). This allows us to associate each Gaussian with text to render the 2D semantic map. Please refer to [5] for more detailed explanations.
>
>
> **Reference:**
> [1]. DUSt3R: Geometric 3D Vision Made Easy. CVPR 2024
> [2]. pixelnerf: Neural radiance fields from one or few images. CVPR 2021
> [3]. pixelSplat: 3D Gaussian Splats from Image Pairs for Scalable Generalizable 3D Reconstruction. CVPR 2024
> [4]. Splatter Image: Ultra-Fast Single-View 3D Reconstruction, CVPR 2024
> [5].  Language-driven Semantic Segmentation, ICLR 2022

---

> > ### Author Response · Authors · 2024-08-14
> > **Respectfully Requesting Comments from the Reviewer**
> >
> > Dear Reviewer K6pm,
> >
> > Thank you once again for your review. As the deadline for the author-reviewer discussion approaches, we noticed that we haven't received any further comments from you.
> >
> > We have addressed all your questions with additional experiments and clarifications:
> > - We have demonstrated the importance and novelty of the problem: "Semantic 3D reconstruction directly from unposed images using a **single differentiable model**."
> > - We provided comparisons showing that our method is not only pose-free but also enables **more 3D tasks** than other methods.
> > - We plan to adjust the title to better emphasize the unification of 3D tasks that our approach offers.
> > - We clarified the necessity of a model that maps global 3D points for input images.
> > - We provided further clarification on how our method compares with Free3D.
> > - We explained the terminology you requested.
> >
> > As the discussion period is coming to an end, we would greatly appreciate any additional feedback you might have. If our responses have clarified your understanding of our paper, we sincerely hope you might consider raising the rating.
> >
> > Thank you again for your effort in reviewing our paper.
> >
> > Best regards,
> > Authors of Paper 3523

---

### Author Rebuttal · Authors · 2024-08-07

We thank all reviewers for acknowledging that the work is sound and clearly presented (8Lhg). The presented Transformer-based design is very valuable (fir7) and general (K6pm), running lighting-fast (K6pm, fir7, MCB6, 8Lhg) while achieving compelling quality (K6pm, 8Lhg). We have addressed all the questions posed by the reviewers with additional experimental results. We will carefully revise our main manuscript, following those suggestions.

---

### Decision · Program_Chairs · 2024-09-25

**Decision:**

Accept (poster)

**Comment:**

This paper proposes the Large Scene Model (LSM) to process unposed RGB images directly into semantic radiance fields. Evaluation is performed on eight unseen datasets from ScanNet, and the results are impressive compared to related methods based on posed RGB images. The paper received unanimous praise from all reviewers for its low computational cost and competitive accuracy on three different tasks: NVS, depth estimation, and semantic segmentation. AC recommends the authors to include all information in the rebuttal into the final camera-ready version and congratulates the authors on the paper acceptance.